# The Importance of Technology in the Combined Interventions of Cognitive Stimulation and Physical Activity in Cognitive Function in the Elderly: A Systematic Review

**DOI:** 10.3390/healthcare11172375

**Published:** 2023-08-23

**Authors:** Rute Rocha, Sara Margarida Fernandes, Isabel M. Santos

**Affiliations:** 1Centro Social e Paroquial de S. Mamede do Coronado, Aveiro University, 3810-193 Aveiro, Portugal; 2Portucalense Institute for Psychology (I2P), Portucalense University, 4200-072 Porto, Portugal; 3William James Center for Research, Department of Education and Psychology, University of Aveiro, 3810-193 Aveiro, Portugal; isabel.santos@ua.pt

**Keywords:** cognitive stimulation, physical activity, combined interventions, older adults, new technologies, exergaming, e-mental health

## Abstract

Background: Numerous studies have been developed in an attempt to understand which factors best predict improvements in cognitive function in the elderly such as exergaming. The aim of this study was to investigate and systematize literature on intervention programs that simultaneously include cognitive stimulation and physical activity, understand the importance of the use of new technology, including exergaming or computer programs, and understand their impact on cognitive function in older adults, giving indications about their contribution to healthy aging. Methods: A narrative approach was used for extraction and synthesis of the data. Relevant studies were identified from electronic databases such as PubMed, Scopus, Web of Science, and Academic Search Complete. Results: Thirty-two articles, involving 2815 participants, were identified. All selected studies were randomized controlled studies. The studies were published between 2011 and 2020. All studies included a combination of cognitive and physical interventions. Many of the studies used technology to administer the cognitive stimulation program. Conclusions: Most of the analyzed studies used exergaming in physical and cognitive interventions, demonstrating that this new form of intervention exerts lasting and stable benefits in cognition. However, we concluded that more studies are needed to compare interventions that use exergaming or computer programs with traditional interventions.

## 1. Introduction

Numerous researchers have attempted to understand which factors best predict improvements in cognitive functions in older adults. Some of these factors are diet, cognitive stimulation, physical activity, sleep quality, among others. Several studies have suggested a significant impact of these factors on general cognitive functioning, especially memory, executive functioning, learning, and attention.

In recent decades, there has been a growing interest in cognitive changes that occur in the aging population and how we can prevent them.

Aging is a continuous and complex process characterized by progressive physical and psychological changes [1]. In the most developed countries, the proportion of older adults has increased significantly, as a result of the decrease in birth rates and the increase in average life expectancy [2]. The increase in demographic aging implies new challenges for society, such as keeping older adults healthy, active, and participative for a longer period of time, as advocated by the World Health Organization [3].

When considering a prescription for healthy and successful aging, the vital role of cognitive stimulation and physical activity is remarkable. This highlights the importance of non-pharmacological interventions, related to changes in the population’s lifestyle [4]. In the last decade, issues associated with new technologies, especially exergaming, have gained prominence and they have been used in the development of non-invasive intervention techniques.

### 1.1. Physical Activity

Numerous benefits of physical exercise are recognized for the health and well-being of the general population. Regular physical activity promotes the improvement of physical and mental functions [5] and is associated with decreased risk of mortality, diabetes, cardiovascular problems, stroke, and breast and colon cancer [6]. In addition, it promotes improvements in bone health, in reducing the risk of osteoporosis, in postural stability, and in increasing flexibility, promoting a decrease in the risk of falls, one of the main causes of mortality in the aging population [7].

Physical activity is also important when it comes to mental health, allowing the prevention of dementia, maintaining independence, improving quality of life [8], and is associated with improvements in the level of depression [9], stress, and anxiety [7]. Physical activity can decrease the risk of dementia by 28% [10], reverse some of the unwanted effects of a sedentary lifestyle, and contribute to delay brain aging and degenerative pathologies, such as Alzheimer’s disease [11], being fundamental in improving memory and learning [12]. In addition, it allows the attenuation of the effects of aging at the brain level, being fundamental for good cognitive functioning [10,13,14].

High levels of physical activity are associated with increased brain activity [15], increasing the brain’s ability to establish new connections. Thus, by allowing higher levels of oxygen to be pumped into the body, it has a very positive impact on brain activity. Physical activity improves neuroplasticity, changing the synaptic structure and function in various brain regions [16]. Physical activity modifies trophic signaling factors and neuronal function and structure in key areas for cognition [17].

### 1.2. Cognitive Stimulation

These cognitive changes are different in each individual; in some cases, they occur suddenly and, in others, in a gradual way [18]. Thus, the growing study of this phenomenon promotes the improvement of diagnostic techniques, capable of distinguishing between normative and pathological processes of aging, and of preventive intervention, capable of outlining the most effective methods for active and meaningful longevity.

The aging process brings losses at the cognitive and functional level [19]. The cognitive abilities that show the greatest deficit in the context of a normative aging process are memory, visuospatial capacity, and information processing speed [20]. According to Calatayud et al. (2018) [19], interventions aimed at exercising and stimulating cognitive abilities may contribute to a reduction in the negative effects of aging, delaying the appearance of deterioration of the various cognitive functions.

Cognitive stimulation refers to a set of methods and techniques that aim to optimize the performance of cognitive functions, through compensation strategies and cognitive reserve, in order to enhance neuroplasticity [19] and it is based on the general view that a lack of cognitive activity accelerates cognitive decline [21]. Cognitive stimulation promotes “involvement in activities aimed at the general improvement of social and cognitive functioning, in order to compensate irremediable neurocognitive deficits and maintain the daily function preserved for as long as possible” [22] (p. 406).

### 1.3. Physical Activity and Cognitive Stimulation Combined: The Role of Exergaming

Physical exercise alone may not be enough to delay cognitive decline in the aging. Research has suggested that combining physical exercise with cognitive stimulation may be a more successful strategy [17,23,24].

Adcock et al. (2020) [16] suggested that we must think about physical and cognitive activity inseparably, since muscles and body movements are controlled by the central nervous system, while feedback from peripheral structures such as muscles and sensory organs influence brain activity. For the same authors, mild cognitive decline and dementia are associated with greater physical decline, compared to older adults with normal cognitive functioning.

Exergaming can be very useful in cognitive stimulation, favoring intervention tasks [25]. According to some studies, the combination of physical activity and cognitive stimulation using exergaming seems to be an effective strategy in the long term. One of these strategies is the use of exergaming. Video game playing may have cognitive benefits and it is highly motivating and likely to promote exercise adherence [26].

Thus, this systematic literature review aimed to investigate and systematize the literature on intervention programs that combined cognitive stimulation and physical activity published in the last decade, giving indications about their efficacy and, additionally, to understand the importance of exergaming or computers programs in the intervention with the elderly.

## 2. Materials and Methods

### 2.1. Protocol and Registration

This work adhered to the Preferred Reporting Items for Systematic Reviews and Meta-Analyses (PRISMA) [27] statement guidelines and was registered on PROSPERO with the number CRD42021231687.

### 2.2. Literature Search Strategy and Study Selection

Articles in English published in electronic databases such as PubMed, Scopus, Web of Science, and Academic Search Complete, from first records to 27 July 2022, were reviewed. The following keywords were used: “cognitive stimulation”, “physical activity”, and “older adults”. No restrictions on language or publication type were applied. Two of the authors independently conducted an initial screening of titles and abstracts and assessed full-text versions of potentially relevant articles. Disagreements were resolved by a third author. The electronic search was complemented by hand-searching the references of included papers and previous reviews.

### 2.3. Eligibility Criteria

Studies were eligible for final inclusion in the systematic review if they cumulatively met the following criteria: studies carried out with older adults and whose participants took part in physical and cognitive intervention programs simultaneously; interventions where cognitive functioning has been evaluated; studies with peer review; randomized controlled studies; and no restrictions on language or publication type were applied at the moment of the database search. In the present study, the following types of publication were excluded: letters to the editor; comments; editorials; systematic reviews; and meta-analyses studies. Finally, studies that did not investigate the effect of the combination of physical activity and cognitive stimulation on the cognition of the older adults were also excluded.

### 2.4. Data Extraction Process

At first, possible studies for inclusion in the systematic review were identified. An electronic data extraction form was used, using the Excel program. After removing the duplicates, the titles and abstracts of the studies were screened. In this way, the relevant studies were extracted, according to the pre-defined inclusion criteria. Studies that did not meet the inclusion criteria were removed. In this study, the extracted data were synthesized in a table, with the following information: authors, year, sample size, frequency of intervention, study design, existence and timing of follow-up, type of cognitive intervention, type of physical intervention, objectives, and main results.

### 2.5. Quality Assessment

The researchers assessed the risk of bias in individual studies using the revised Cochrane risk of bias tool for randomized trials (RBO2) [28].

## 3. Results

### 3.1. Selection of Studies

As shown in Figure 1, a total of 11,096 studies were identified through electronic database searching and one additional article was identified by back citation. After removing 555 duplicate records, 10,542 studies were screened based upon title and abstract, but 10,426 did not meet the eligibility criteria. The full text was retrieved for 116 articles, of which 84 were excluded.

Thirty-two studies were identified that combined physical activity and cognitive stimulation. Of the thirty-two analyzed, eighteen studies (56.3%) presented a low risk of bias. Five (15.6%) studies had a high risk of bias. Finally, nine studies (28.1%) had some concerns (Table 1).

### 3.2. Type of Studies

Studies published in peer-reviewed journals were selected. Of the studies analyzed, eleven conducted a follow-up test after their interventions. Follow-up varied between 3 months [29,30,31,32], 6 months [33,34], 1 year [35], and 2 years [24]. One of the analyzed studies carried out two follow-up moments, at 3 months and at 6 months [38] (Table 2).

### 3.3. Type of Participants

Most studies included people over the age of 60. Participants with mild cognitive impairment, with dementia, and who were cognitively healthy were included (Table 2).

Regarding the recruitment of the sample, participants were recruited from community centers [23,26,39,40], day care centers [38], residential structures for the elderly [35,41,42,43,44,45], in the community in general [16,29,34,35,36,37,46,47,48,49,50], senior organizations [16], and in hospitals and neurology clinics, normally associated with universities [24,30,32,33,51,52]. Participants in thirteen studies were diagnosed with dementia or mild cognitive impairment.

### 3.4. Duration of the Interventions

Regarding the duration of the interventions, the studies varied between 7 weeks [53] and 40 weeks [54], with physical and cognitive training that varied between 10 and 110 min per session, which were divided by two to five days a week.

### 3.5. Intervention Groups

Several studies did not have a control group [23,36,37,40,41], and they compared the effectiveness of physical activity and cognitive stimulation with each of its components administered in isolation. All other studies compared the combination of physical activity and cognitive stimulation with the isolated administration of cognitive stimulation or a control group, in the cognitive functioning of the older people.

### 3.6. Assessment of Participants

Regarding the neuropsychological assessment tests used, most studies used the Mini Mental State Examination (MMSE) to screen the sample, and participants should fall within a range of scores. Only six studies did not use the MMSE as a cognitive screening test [16,23,36,40,46,49]. In all studies, participants underwent an extensive battery of neuropsychological tests to assess various cognitive domains.

### 3.7. Type of Interventions

Studies were included if they focused on interventions simultaneously combining cognitive stimulation with physical activity in older adults. Cognitive stimulation was focused on several domains of cognitive functioning, for example, memory, attention, executive functions, spatial-temporal activities, language, and others. The intervention could include structured cognitive intervention programs [23,49]. Physical exercises included any form of structured physical activity, such as aerobic exercise, strength or functional, toning, cycling, and walking.

### 3.8. Use of Technologies

Some of the studies in this review used exergaming or computer programs (43%). Exergaming interventions were included because the gaming activities combined physical activity with cognitive stimulation [22,35,38] (Table 2).

**Table 2 healthcare-11-02375-t002:** Summary of studies involving physical activity and cognitive stimulation.

Authors and Year	n	Interventions Frequency	Follow-Up	Cognitive Intervention Component	Comparison Groups	Physical Intervention Component	Participant Type	Aims	Major Findings
Adcock et al., (2020) [16]	37	30–40 min; 3×/week 16 weeks	No	Step-based cognitive exercises	Passive control	Tai-chi, dancing, step training (exergame)	Cognitively health	Evaluate the effect of an exergaming workout, performed at home, on physical activity, cognitive functioning, and brain volume.	The findings indicated a positive influence of exergame training on executive functioning. No improvements in physical functions or brain volume were evident in this study.
Anderson-Hanley et al., (2018) [51]	13	45 min; 4×/week 24 week	No	Exergame	PA	Cycling	Mild Cognitive Impairment	The Aerobic and Cognitive Exercise Study (ACES) sought to replicate and extend prior findings of added cognitive benefit from exergaming to those with or at risk for mild cognitive impairment (MCI).	There were significant moderate improvements in executive functions and verbal memory. Effects appeared to generalize to self-reported everyday cognitive function.
Bacha et al., (2018) [53]	46	60 min; 2×/week; 7 weeks	No	Exergame	PA	Exergame	Cognitively health	To compare the effectiveness of Kinect Adventures games versus conventional physiotherapy to improve postural control, gait, cardiorespiratory fitness, and cognition of the elderly.	Both interventions provided positive effects on postural control, gait, cardiorespiratory fitness, and cognition of the elderly.
Barban et al., (2017) [30]	481	CT = 30 min; 2×/week PA = 30 min; 2×/week 12 weeks	Yes (3 months)	Multidomain CCT-Executive function	CT	Motor training	Cognitively health	Study the impact of physical activity and cognitive stimulation on fear of falling	The results demonstrated significant improvements in executive functions.
Barcelos et al., (2015) [53]	17	20–45 min; 2–5×/week 12 weeks		Exergaming–Executive function	PA	Cycling	Cognitively health	Investigate whether greater cognitive challenge while exergaming would yield differential outcomes in executive function and generalize to everyday functioning.	Pilot data indicated that for older adults, cognitive benefit while exergaming increased concomitantly with higher doses of interactive mental challenge.
Barnes et al., (2013) [46]	126	1 h; 3×/week 12 weeks	No	Multidomain	PA, CT and CCT	Aerobic exercise and strength exercise	Mild Cognitive Impairment	Examine the combined effects of physical plus mental activity on cognitive function.	In inactive older adults with cognitive complaints, physical and mental activity was associated with significant improvements in global cognitive function.
Boa Sorte Silva et al., (2018) [47]	127	60 min; 3×/week 24 weeks	No	Square-stepping exercise, memorizing complex stepping patterns	PA	Aerobic, stepping training and resistance	Mild Cognitive Impairment	Investigated the effects of multiple-modality exercise with additional mind-motor training on cognition in community-dwelling older adults with subjective cognitive complaints.	Additional mind-motor training did not impart greater immediate benefits to cognition among the study participants.
Desjardins-Crépeau et al., (2016) [48]	76	12 weeks; 24 sessions of 60 min of physical exercise 12 sessions of 60 min of cognitive stimulation 3×/week (2 physical and 1 cognitive)	No	Dual Task Exercises	CT, PA and CCT	Treadmill walking and resistance training	Cognitively healthy	Examine the effects of combined physical and cognitive interventions on physical fitness and neuropsychological performance in healthy elderly people.	There were no significant improvements.
Donnezan et al., (2018) [33]	69	24 sessions lasting 1 h over 12 weeks 2×/week	Yes (six months)	Executive functions, working memory, namely, mental flexibility, inhibition, reasoning and updating.	Passive control	Aerobic training on bikes,	Mild Cognitive Impairment	Compare the benefits of cognitive and physical training simultaneously, with each training administered separately in executive, cardiorespiratory and walking functions, to assess a potential additional additive effect.	It was more advantageous to administer cognitive training and physical activity simultaneously than alone.
Eggenberger et al., (2015) [35]	89	1 h; 2×/week 6 months	Yes (1 year)	Verbal memory training	PA	Dance, walk, strength and balance exercises	Mild Cognitive Impairment	Understand if the combination of physical activity and cognitive stimulation has greater benefits for the elderly, compared to physical activity practiced in isolation.	Executive functions benefited from simultaneous cognitive–physical training compared to exclusively physical multicomponent training.
Eggenberger et al., (2016) [56]	33	30 min; 3×/week 8 weeks	No	Videogame dancing	PA	Videogame dancing, with exergaming	Cognitively healthy	This study aimed to investigate if exercise training induces functional brain plasticity during challenging treadmill walking and elicits associated changes in cognitive executive functions.	There were improvements in executive functioning.
González-Palau et al., (2014) [23]	50	1h of physical training 40 min of cognitive stimulation 3×/week 12 weeks	No	Attention, perception, episodic memory and working memory.	PA and CT	Warm-up, aerobic exercises, endurance, strength, balance, stretching and cool-down training exercises.	Cognitively healthy and Mild Cognitive Impairment	The main objective of this study was to present the preliminary results that determine the possible effectiveness of the Long Lasting Memories program in the improvement of cognitive functions and symptoms of depression in healthy elderly and subjects with mild cognitive impairment.	Significant improvements after the implementation of the program.
Htut et al., (2018) [42]	84	30 min; 3×/week 8 weeks	No	X-box 360 games (exergaming)	Passive control	X-box 360 games (exergaming)	Cognitively healthy	Compared the effects of Physical exercise, virtual reality-based exercise, and brain exercise on balance, muscle strength, cognition, and fall concern.	Significant improvements in checking physical and cognition
Karssemeijer et al., (2019) [24]	115	12 weeks, 30–50 min; 3×/week.	Yes (12 and 24 months)	Executive functions, episodic memory, working memory, psychomotor speed. (exergaming)	PA	Aerobic training, relaxation, and flexibility exercises.	Dementia	Investigate the effect of training with exergame and aerobic training on cognitive functioning in elderly people with dementia.	Significant improvements were found in the combination of physical activity and cognitive stimulation.
Laatar et al., (2018) [34]	24	60 min; 3×/week 24 week	Yes (6 months)	Cognitive tasks	PA	Balance and strength exercises	Cognitively healthy	This study examined postural, physical and cognitive performances and postural performance during daily life tasks in older adults pre- and post-6 months physical and physical-cognitive interventions.	Only simultaneous physical-cognitive training modality enhanced performance in some tasks relative to everyday abilities. Nonetheless, these gains were lost after 3 months of detraining period suggesting a need for older people to participate regularly in such training for their daily life independence.
Legault et al., (2011) [49]	67	CT = 10–12 min; 2×/week 2 months PA = 150 min/week 4 months	No	Memory, executive functions	CT, CCT, PA	Aerobic and flexibility	Cognitively healthy	Understand the effectiveness of a physical and cognitive intervention program on the cognitive functioning of the elderly.	The interventions produced marked improvements in cognitive and physical performance measures.
Linde and Alfermann (2014) [29]	55	40 min 16 weeks	Yes (12 weeks)	Information processing speed, short-term memory, spatial relations, concentration, reasoning, and cognitive speed	Passive control	Aerobic and strength	Cognitively healthy	The objective of this study was to analyze the short- and long-term effects of physical, cognitive, and combined physical plus cognitive training regimens on age-sensitive fluid cognitive abilities.	Physical, cognitive, and combined physical plus cognitive activity can be seen as cognition-enrichment behaviors in healthy older adults that showed different rather than equal intervention effects.
Maci et al., (2012) [52]	14	3 months; 1 h of physical activity, 1 h of cognitive stimulation and 30 min of group discussion; 5×/week.	No	Spatiotemporal orientation, memory, executive skills, and language.	Passive control	Aerobic exercise of mild intensity.	Dementia	Evaluate the effect of cognitive stimulation, physical activity, and socialization in patients with AD and on the quality of life and mood of their informal caregivers.	Significant improvements were found in the combination of physical activity and cognitive stimulation.
Maffei et al., (2017) [37]	113	CT = 60 min; 3×/week 7 months PA = 1hour; 3×/week 7 months	Yes (1 year)	Multidomain	Passive control	Aerobic, strength and flexibility	Mild Cognitive Impairment	Assess the efficacy of combined physical-cognitive training on cognitive decline, Gray Matter (GM) volume loss and Cerebral Blood Flow (CBF) in hippocampus and parahippocampal areas, and on brain-blood-oxygenation level- dependent (BOLD) activity elicited by a cognitive task	The results showed that a non-pharmacological, multicomponent intervention improved cognitive status and indicators of brain health in MCI subjects.
Maillot et al., (2012) [26]	32	60 min; 2×/week 12 weeks	No	Nintendo Wii games (exergaming)	Passive control	Nintendo Wii games (exergaming)	Cognitively healthy	The purpose of this study was to assess the potential of exergame training based on physically simulated sport play as a mode of physical activity that could have cognitive. benefits for older adults.	The trainees improved significantly in measures of game performance. They also improved significantly more than the control participants in measures of physical function and cognitive measures of executive control and processing speed functions, but not on visuospatial measures.
McDaniel et al., (2014) [39]	79	60 min; 6×/week (CT = 3; PA = 3) 24 weeks (PA = 24; CT = 8)	No	Attentional coordination, prospective memory, and retrospective-memory retrieval	Passive control	Aerobic Exercise	Cognitively healthy	Investigate the potential benefits of a novel cognitive training protocol and an aerobic exercise intervention, both individually and in concert, on older adults’ performances in laboratory simulations of select real-world tasks.	The findings suggested that at least for everyday oriented prospective memory tasks involving cognitive challenges, well-designed cognitive training programs may confer more robust gains in performance than a standard aerobic exercise program over a limited (6-month) period.
Nishiguchi et al., (2015) [43]	48	90 min; 1×/week 12 weeks	No	Multidomain	Passive control	Walking, strength	Cognitively healthy	Investigate whether a 12-week physical and cognitive exercise program can improve cognitive function and brain activation efficiency in community dwelling older adults.	Physical activity alone and combined cognitive and exercise training can improve cognitive function in older adults.
Norouzi et al., (2019) [31]	60	60–80 min; 3×/week 4 weeks	Yes (12 weeks)	Cognitive tasks	PA	Resistance	Cognitively healthy	The aim of this study was to investigate whether and to what extent two different dual-task interventions improved both working memory and balancing.	Dual-task interventions improved both balance performance and working memory, but more so if cognitive performance was specifically trained along with resistance training
Park et al., (2019) [21]	49	110 min 24 weeks	Yes (3 months)	Word games, memory, numerical calculations	Passive control	Aerobic exercise (stair stepping, walking and stair climbing) warm-up, stretching, balance exercise,	Mild Cognitive Impairment	Investigate the association between a dual-task intervention program and cognitive and physical functions.	The 24-week combined intervention improved cognitive function and physical function in patients with MCI relative to controls.
Rahe et al., (2015b) [40]	30	90 min; 2×/week 6.5 weeks	Yes (1 year)	Memory, attention, and cognitive functions	CT	Strength, flexibility, and coordination/balance	Cognitively healthy	Compare the effect of the combination of cognitive stimulation and physical activity with the administration of cognitive stimulation separately, in healthy elderly people.	The results suggested that the combination of physical activity and cognitive stimulation is advantageous in terms of long-term care.
Rahe et al., (2015) [36]	68	90 min; 2×/week 7 or 8 weeks;	No	Multidomain and/or counseling	CT	Strength, flexibility, coordination, endurance, and aerobic exercise	Cognitively healthy	Understand the additional benefit of combining physical activity with cognitive stimulation in healthy aging.	Data was inconsistent concerning the question of whether cognitive-physical training yields stronger cognitive gains than cognitive training.
Rezola-Pardo et al. (2019) [44]	85	60 min; 2×/week 12 weeks;	No	Cognitive tasks (Attention, executive function, semantic memory)	PA	Strength and balance	Cognitively healthy	Determine whether the addition of simultaneous cognitive training to a multicomponent exercise program offers further benefits to dual-task, physical and cognitive performance, psycho-affective status, quality of life and frailty in older adults living in long-term nursing homes.	The addition of simultaneous cognitive training did not seem to offer significantly greater benefits to the evaluated multicomponent exercise program in older adults.
Shatil, (2013) [50]	122	CT = 40 min; 3×/week; 16 weeks; PA = 45 min; 3×/week; 16 weeks	No	Multidomain	CT and PA	Aerobic exercise, strength, flexibility, aerobic warm-up, cardiovascular workout seated and standing, aerobic cool-down	Cognitively healthy	Understand the effect of the combination of physical activity and cognitive stimulation and compare its effect with each of the interventions administered in isolation.	The results suggested that the elderly who were submitted to cognitive stimulation had more benefits in terms of cognitive functioning.
Shimada et al., (2018) [54]	308	90 min; 1×/week 40 weeks	No	Multidomain CT	Health education control group	Aerobic, strength, and balance	Mild Cognitive Impairment	To compare the cognitive and mobility effects of a 40-week program of combined cognitive and physical activity with those of a health education program.	Combined physical and cognitive activity improved or maintained cognitive and physical performance in older adults with mild cognitive impairment, especially the amnestic type.
Van het Reve and de Bruin (2014) [45]	156	CT = 10 min; 3×/week; PA = 40 min; 2×/week; 12 weeks;	No	Single domain CCT (attention)	PA	Strength and balance	Cognitively healthy	To assess the effects of the combination of physical activity and cognitive stimulation on walking and cognitive functioning.	Combining strength-balance training with specific cognitive training had a positive additional effect on dual task costs of walking, gait initiation, and divided attention.
Van Santen et al., (2020) [38]	112	2×/week; 6 months;	Yes (3 and 6 months	Multidomain (exergaming)	PA and CT	Cycling	Dementia	Evaluate the effects of exergaming in the cognition.	Mixed-model analyses showed no statistically significant effects on primary outcomes
Yoon et al., (2013) [41]	30	30 min; 3×/week 12 weeks	No	Memory	CTE and CT	Cycling exercise	Dementia	Understand the additional benefit of combining physical activity with cognitive stimulation in healthy aging.	Significant improvements were found in the combination of physical activity and cognitive stimulation.

Note. CCT = computerized cognitive training; CT = cognitive training; PA = physical activity; RCT = randomized control trial; AD = Alzheimer’s disease; CTE = cognitive training combined with physical activity.

### 3.9. Results of the Combination of Interventions on Cognitive Functioning

The analysis of the studies demonstrated that there was a significant impact of the combination of cognitive stimulation and physical activity on the cognitive functioning of elderly people with different cognitive performances. According to the analysis of the results, executive functioning is the cognitive function where the greatest impact of the combination of these two types of intervention is observed [16,30,35,56].

In inactive older adults with cognitive complaints, physical and mental activity was associated with significant improvements in global cognitive function [46] in the long term [40]. It was more advantageous to administer cognitive training and physical activity simultaneously than isolated [33].

## 4. Discussion

The objectives of this systematic review of the literature were to systematize all recent studies on the combined interventions of physical activity and cognitive stimulation on the cognitive functioning of older adults, to understand the impact of the use of exergaming integrated in these interventions, and to obtain information about its effectiveness in contributing to healthy aging.

Many of the studies used technology to administer the cognitive stimulation program, namely exergaming or computer programs. Several studies suggest that the use of technology, namely computer programs, can be advantageous for the cognitive functioning of the older adults, namely in terms of memory [57]. Exergame training seems to be a motivating and promising option for simultaneous physical–cognitive training in older adults [22]. The use of these programs is expected to allow the prevention and treatment of cognitive decline, to decrease a sedentary lifestyle, cause a reduction in the time spent on the intervention, and cause a decrease in the necessary resources [23]. This type of intervention seemed to be more effective in treating the elderly compared to traditional interventions. They can be used in cognitive stimulation and physical activity. Most of the studies used the exergaming technique, which consisted of activities that included physical activity and cognitive stimulation in an articulated way and simultaneously. One of the most well-known and publicized instruments is the Wii-Fit Pro, which has been widely used in intervention with the elderly. Exergame works as a new health strategy with the purpose of providing cognitive and functional improvement and/or maintenance in the elderly, promoting healthy aging. In fact, all analyzed studies using exergaming demonstrated significant results in improving cognitive functioning. However, the analyzed studies did not compare computerized interventions with traditional interventions, and it was not possible to deduce the difference in effectiveness between the two.

Independently of the use of technology to administer the intervention, most of the studies presented demonstrated that the combination of physical activity and cognitive stimulation is most beneficial for cognitive functioning in the aging population.

However, although this efficacy has been demonstrated, some studies revealed some differences and contradictions [36,44,48]. In two studies [36,48], no significant advantages were found in the combination of physical activity with cognitive stimulation in the aging. In one study [44], inconsistent results were found. These differences can be explained by methodological differences.

Several studies did not have a control group, namely the studies [23,36,37,40,41], although all were randomized control trials. This is perhaps because these studies had the aim of identifying the additional benefit of adding a physical exercise intervention to a cognitive one. For this reason, they compared the results of combining the two interventions with a group that only received cognitive stimulation. For the authors, a control group was considered to be a group that did not benefit from any type of intervention.

Although several intervention programs used physical activity and cognitive stimulation, few investigations analyzed and compared the combination of both, with each of the components administered in isolation and with a control group [24,48,50]. The remaining studies compared the combination of physical activity and cognitive stimulation with only one of the variables in isolation. However, the results were inconclusive [24], significant differences in psychomotor speed were found in the group that received aerobic and cognitive training (exergame) when compared to the control group. However, in same studies, there were no statistically significant differences, since, according to the authors, combined training did not produce synergistic effects [48,50].

In addition, the short-term effects of the interventions did not reveal significant differences in most of the analyzed cognitive functions. Although the discovery that the combination of cognitive stimulation and physical activity significantly improves psychomotor speed may be clinically relevant, as psychomotor speed is an important predictor of functional decline [24], more conclusive studies are needed regarding the effect of physical activity and cognitive stimulation on different cognitive functions. Some of these cognitive functions are less sensitive to the intervention of physical activity, with no significant changes in these cases, which is why studies in this regard remain fundamental [48].

Through the results obtained, it is possible to verify that thirteen studies were carried out with a sample with mild cognitive impairment or a formal diagnosis of dementia. These results suggest that there was some variety in the studies developed, with them not being focused only on a segment of the population. The fact that, in recent years, numerous studies have been carried out with cognitively healthy subjects is advantageous, since it allows the development of increasingly effective strategies to promote healthy aging.

Only eleven studies performed a follow-up after their interventions. This is a limitation that some authors identified [58] and that we agree with, since it does not allow drawing conclusions about the effectiveness of the combination of physical activity and long-term cognitive stimulation. Studies with follow-ups are sometimes inconclusive, not revealing any significant changes over the long term in most conditions analyzed [24].

Based on the analysis of the studies, it is possible to conclude that the factors that influence cognitive and physical dysfunction are susceptible to intervention. Thus, neuropsychological and physical interventions can be fundamental for normative aging, since they have the potential to delay the decline in cognitive and physical functions. Given the cognitive improvements, after the interventions, it is possible to infer that brain plasticity is present in the aging population and that cognitive decline can be delayed in cognitively healthy participants and individuals with mild cognitive impairment [23].

### Limitations and Future Directions

Several key limitations of this comprehensive systematic review should be acknowledged. The characteristics of the studies included in the analysis varied widely, for example, in the focus of the intervention, in the objectives, in the number and time of the sessions, in the sample size, in the cognitive functions evaluated, and in the type of physical activity implemented. For this reason, meta-analysis or statistical comparisons were not possible.

Future studies should focus on the development of new technologies, applying them in their intervention programs, in an attempt to make them more effective; on the other hand, future studies should: compare the effectiveness of interventions that use exergaming with traditional interventions; use follow-up strategies to verify the effects of this type of intervention in the long term; understand the impact of the combination of the two forms of intervention on other personal individual variables (depression, anxiety, quality of life, psychological well-being and performance, daily living activities); compare the effects of the combination of physical activity and cognitive stimulation with each of its components administered separately; and, finally, future studies should try to distinguish the benefits of the combination of physical activity and cognitive stimulation according to the user’s cognitive level (cognitively healthy, mild cognitive impairment, and dementia).

We also found that the various studies analyzed only addressed the effects of the intervention on cognition, not mentioning the positive impact that cognitive stimulation may have on the performance of activities of daily living for the elderly. Several studies have addressed the issue of the transfer effect, claiming that cognitive stimulation does not generalize beyond the specific tasks that are worked on [59]. The concept of transfer consists in the generalization of the results obtained to other domains of cognition or to tasks that, in some way, differ from the target of the training program [59]. Therefore, future studies should seek to understand the effect of the combination of cognitive stimulation and physical activity in the performance of daily living activities in the elderly.

## 5. Conclusions

As scientific research in the area of aging progresses, the results suggest that a healthy lifestyle, with regular practice of physical activity and constant investment in cognitive stimulation activity, is essential for a healthy and meaningful process of aging. It was more advantageous to administer cognitive training and physical activity simultaneously than alone [33]. Combining physical exercise with cognitive stimulation is a more successful strategy [25]. Both cognitive stimulation and physical training induce changes in the brain function and structure of healthy elderly people, namely in the functioning of the frontal lobe and in the increase in white matter in the frontal and parietal regions [58].

Despite not finding studies that compared the effectiveness of using traditional stimulation techniques with those using exergaming, these seem to have advantages compared to the rest [57]. The use of new technology promotes, in addition to the significant improvement in cognition, the reduction in sedentary lifestyles, the reduction in time spent in the intervention, and the reduction in the necessary resources.

Recent studies indicate that cognitive activities are complex and challenging, mobilizing more than one cognitive domain simultaneously. In the coming decades, we must join efforts to associate new technologies with cognitive stimulation activities, in order to make our interventions increasingly effective. On the other hand, future studies could be carried out to assess the needs of the elderly, with regard to technologies, specifically in terms of motivators and barriers to their use.

## Figures and Tables

**Figure 1 healthcare-11-02375-f001:**
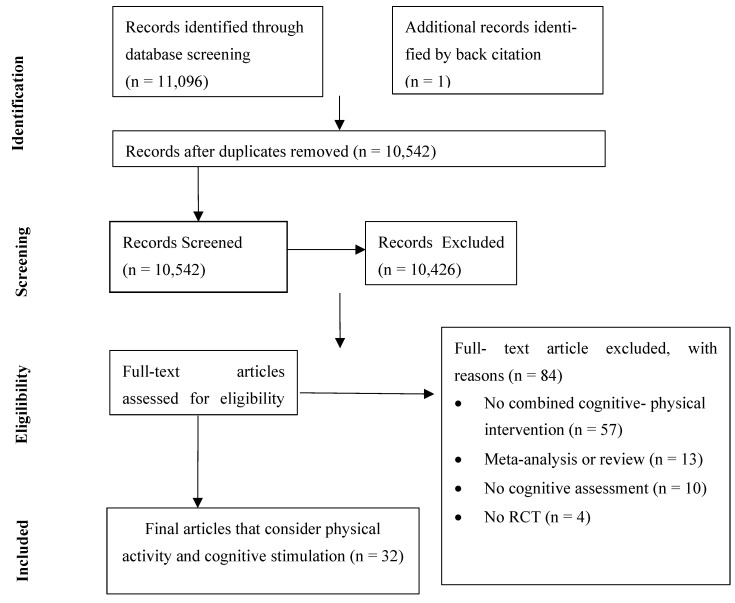
Flowchart of the selection process of studies for inclusion in the systematic review.

**Table 1 healthcare-11-02375-t001:** Critical appraisal of the studies with randomized design included in the review, based on Cochrane risk of bias tool.

Study	Random Sequence Generation	Allocation Concealment	Blinding of Patients, Personnel	Incomplete Outcome Data	Selective Outcome Reporting	Overall Bias
[16]	Low	Low	High	Some concerns	Low	Some concerns
[23]	Some concerns	High	High	Some concern	Some Concern	High
[24]	Low	Some concerns	Some concerns	Low	Low	Low
[26]	Some concern	Some concern	Low	Some concern	Some concern	Some concern
[29]	Low	Some concern	Some concern	Some concern	Some concern	Some concern
[30]	Low	Some concerns	Some concerns	Low	Low	Low
[31]	Low	Low	Low	Low	Low	Low
[32]	Some concerns	Some concerns	Some concerns	Low	Some concerns	Some concerns
[33]	Some concerns	High	High	Some concerns	Some Concerns	High
[34]	Low	Low	Low	Low	Low	Low
[35]	Some concerns	Some concerns	High	Some concerns	Some concerns	Some concerns
[36]	Low	Some concerns	Some concerns	Low	Low	Low
[37]	Low	Low	Some concerns	Low	Low	Low
[38]	Some concerns	Some concerns	High	Some concerns	Some concerns	Some concerns
[39]	Low	Some concern	Low	Low	Low	Low
[40]	Low	Low	High	Low	Low	Low
[41]	Low	Some concerns	Some concerns	High	High	High
[42]	Some concerns	Low	Low	Low	Low	Low
[43]	Low	Some concerns	Low	Low	Some concerns	Low
[44]	Some concern	Some concern	Some concern	Low	Low	Some concern
[45]	Low	Some concern	Low	Low	Low	Low
[46]	Low	Low	Low	Low	Low	Low
[47]	Low	Low	Low	Some concern	Low	Low
[48]	Some concerns	High	Some concerns	High	Some concerns	High
[49]	Some concerns	Some concerns	Some concerns	Low	Low	Some concerns
[50]	Some concerns	Some concerns	Some concerns	High	High	High
[51]	Low	Some concern	Low	Low	Low	Low
[52]	Low	Some concerns	Low	Low	Some concerns	Low
[53]	Low	Some concern	Low	Low	Low	Low
[54]	Low	Low	Low	Low	Low	Low
[55]	Low	Low	Low	Low	Low	Low
[56]	Low	Some concern	High	Some concern	Low	Some concern

## Data Availability

No new data were created or analyzed in this study. Data sharing is not applicable to this article.

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
