# Peer review of "The Importance of Technology in the Combined Interventions of Cognitive Stimulation and Physical Activity in Cognitive Function in the Elderly: A Systematic Review"

_healthcare, 2023, doi:10.3390/healthcare11172375_

Round 1
Reviewer 1 Report
This study aims to systematically review the literature on the cognitive effects of a combination of exercise and cognitive stimulation intervention, and the importance of new technologies in the intervention. The manuscript has several issues that need to be addressed.
1. As one of the aims stated, the authors should report how the combined intervention affects cognitive function in the results, as well as in the conclusion (effectiveness). It is also expected to see the percentage of the number of articles using the new technologies.
2. Authors should also specify the cognitive function as the outcome in the title, and also in the eligibility criteria in the Methods.
3. I would also suggest the authors focus on cognitive function, not healthy aging, in the first sentence of the abstract and from the very beginning of the introduction.
4. Lines 105-121, “2. Materials and methods…” and lines 157-160, “4. Discussion…” should be deleted.
5. I didn’t see any data or results to support the statement of “demonstrating that this new form of intervention exerts lasting and stable benefits in 25 cognition” in lines 25-26 and “the new technologies seem to have advantages compared to the rest” in lines 321-322.
6. In Lines 287-288, the sentence needs correction.
7. Title: “Stud” should be “Study”.
Author Response
Dear Reviewer,
We would like to thank all the comments made to our article entitled "Importance of technology in the combined interventions of cognitive stimulation and physical activity in cognitive function in the elderly: A Systematic Review Study", by Rute Rocha, Sara M. Fernandes and Isabel M .Santos. We make every effort to try to improve our manuscript.
Comments and Responses
- As one of the aims stated, the authors should report how the combined intervention affects cognitive function in the results, as well as in the conclusion (effectiveness). It is also expected to see the percentage of the number of articles using the new technologies.
We thanks for this suggestion. We have added the following information:
On Pag. 7
3.7. “Use of technologies
Some of the studies in this review used new technologies (43%), with exergaming being the most predominant. Exergaming interventions were included because the gaming activities combined physical activity with cognitive stimulation [22, 35, 38] (Table 2).”
On page 7 the following paragraph was added:
3.8 Results of the combination of interventions on cognitive functioning
The analysis of the studies demonstrates that there is a significant impact of the combination of cognitive stimulation and physical activity on the cognitive functioning of elderly people with different cognitive performances. According to the analysis of the results, executive functioning is the cognitive function where the greatest impact of the combination of these two types of intervention is observed [16, 30, 35, 56].
In inactive older adults with cognitive complaints, physical and mental activity was associated with significant improvements in global cognitive function [46], in a long-term [40].
It was more advantageous to administer cognitive training and physical activity simultaneously than alone [33].
- Authors should also specify the cognitive function as the outcome in the title, and also in the eligibility criteria in the Methods.
We thanks for this suggestion.
The title has been changed to read as follows: “Importance of technology in the combined interventions of cognitive stimulation and physical activity in cognitive function in the elderly: A Systematic Review Study”
The following information was added to the eligibility criteria: “interventions where cognitive functioning has been evaluated” (page 4).
- I would also suggest the authors focus on cognitive function, not healthy aging, in the first sentence of the abstract and from the very beginning of the introduction.
We thanks for this suggestion and in order to respond to this suggestion, changes were made to the abstract and introduction. Thus, in the summary, some information was eliminated and other information considered relevant was added, starting as follows:
“Abstract: Numerous studies have been developed in an attempt to understand which factors best predict improvements in cognitive function in the elderly such as exergaming. The aim of this study was to investigate and systematize literature on intervention programs that simultaneously include cognitive stimulation and physical activity, understand the importance of the use of new technology, including exergaming or computer programs, and understand their impact on cognitive function in older adults, giving indications about their contribution to healthy aging. (page 1).
“Introduction
Numerous researchers have attempted to understand which factors best predict improvements in cognitive functions in older adults. Some of these factors are diet, cognitive stimulation, physical activity, sleep quality, among others. Several studies have suggested a significant impact of these factors on general cognitive functioning, especially memory, executive functioning, learning and attention.
In recent decades, there has been a growing interest in cognitive changes that occur in the aging population and how we can prevent them.”(page 1).
- Lines 105-121, “2. Materials and methods…” and lines 157-160, “4. Discussion…” should be deleted.
Thanks for the repair. The information on those lines has been deleted, as suggested. This information was the journal's guidelines for this section and does not form part of the manuscript's content.
- I didn’t see any data or results to support the statement of “demonstrating that this new form of intervention exerts lasting and stable benefits in 25 cognition” in lines 25-26 and “the new technologies seem to have advantages compared to the rest” in lines 321-322.
We thanks for this suggestion, which we have endeavored to respond to.
This information appears in the results of the evaluated studies, whose analysis is shown in table 2, particularly in the study carried out by Adcock et al. (2020). Also Flak et al. (2014) affirms the importance of computer programs in cognition, and we add this reference (page 16). However, it is true that, as we concluded, more studies are needed to compare interventions that use exergaming or computer programs with traditional interventions.
Therefore, this information was added to the abstract : Conclusions: Most of the analyzed studies use exergaming in physical and cognitive interventions, demonstrating that this new form of intervention exerts lasting and stable benefits in cognition. However, we conclude that more studies are needed to compare interventions that use exergaming or computer programs with traditional interventions.
- In Lines 287-288, the sentence needs correction.
Thanks for this suggestion. The sentence has been changed to:
Only eleven studies performed follow-up after their interventions. This is a limitation that some authors identify [58] and that we agree with, since it does not allow drawing conclusions about the effectiveness of the combination of physical activity and long-term cognitive stimulation.
- Title: “Stud” should be “Study”.
Thanks very much for this repair. The word Study in the title was corrected.
Thank you very much in advance for considering our paper for re-evaluation and I remain available to answer any further comments or concerns.
Sincerely,
Rute Rocha

Reviewer 2 Report
Review
This article provides a systematic review on the combination of physical and cognitive interventions as a means to promote cognitive health in older adults. The result is that such interventions are generally successful. The roles of physical and cognitive interventions have been explored in other reviews but this review on the combination of both types of interventions together is a novel combination to the literature. The search, selection and reporting of individual studies seemed strong to me, although I do not have systematic review experience myself. My primary concern with the article is that it does little to evaluate the studies and the reader is left to search for patterns themselves – I may have missed it but is it stated anywhere how many articles found benefits of the combination above each intervention independently/control conditions? Given that the core search seems to be conducted well, I feel that a major revision may be able to resolve the issues outlined below.
Major issues
The authors need to draw conclusions unique to the original aspect of the study – does the combination of interventions lead to more success than either intervention alone? It would be useful to provide a table or paragraph specifically summarizing the findings related to various relevant comparisons (e.g., combined vs cognitive only, combined vs physical only, combined vs wait list controls).
There is a lot of evidence that cognitive training does not work as the effects do not transfer to different tasks. Evaluations of all of the studies should include reference to transfer effects (or consideration of transfer effects). Transfer effect issues should be covered in the introduction: e.g.,
Simons, D. J., Boot, W. R., Charness, N., Gathercole, S. E., Chabris, C. F., Hambrick, D. Z., & Stine-Morrow, E. A. (2016). Do “brain-training” programs work?. Psychological science in the public interest, 17(3), 103-186.
Control group information should be included in Table 2
Minor issues
I was not sold on the technological aspect of the study as a broad concept; for example, there has been a lot of recent work on step-counting devices etc. which was not covered. Perhaps the term technology could be replaced with exergaming?
Final letters of the title are cropped off
Inclusion of journal rules as part of the introduction were probably not supposed to be included (lines 107+).
Author Response
Dear Reviewer,
We would like to thank all the comments made to our article entitled "Importance of technology in the combined interventions of cognitive stimulation and physical activity in cognitive function in the elderly: A Systematic Review Study", by Rute Rocha, Sara M. Fernandes and Isabel M .Santos. We make every effort to try to improve our manuscript.
Comments and Responses
- The authors need to draw conclusions unique to the original aspect of the study – does the combination of interventions lead to more success than either intervention alone? It would be useful to provide a table or paragraph specifically summarizing the findings related to various relevant comparisons (e.g., combined vs cognitive only, combined vs physical only, combined vs wait list controls).
We thank for this suggestion. In order to clarify this issue, the following paragraph was added to the results section (page 7):
“3.8 Results of the combination of interventions on cognitive functioning
The analysis of the studies demonstrates that there is a significant impact of the combination of cognitive stimulation and physical activity on the cognitive functioning of elderly people with different cognitive performances. According to the analysis of the results, executive functioning is the cognitive function where the greatest impact of the combination of these two types of intervention is observed[16, 30, 35, 56].
In inactive older adults with cognitive complaints, physical and mental activity was associated with significant improvements in global cognitive function [46], in a long-term [40]. It was more advantageous to administer cognitive training and physical activity simultaneously than isolated [33].”
- There is a lot of evidence that cognitive training does not work as the effects do not transfer to different tasks. Evaluations of all of the studies should include reference to transfer effects (or consideration of transfer effects). Transfer effect issues should be covered in the introduction: e.g., Simons, D. J., Boot, W. R., Charness, N., Gathercole, S. E., Chabris, C. F., Hambrick, D. Z., & Stine-Morrow, E. A. (2016). Do “brain-training” programs work?. Psychological science in the public interest, 17(3), 103-186.
We thank for this suggestion. The question raised is really important and was a reason for reflection by the team. In this sense, the suggested reference was analyzed and the following paragraph was added in the discussion section (page 16).
“We also found that the various studies analyzed only address the effects of the in-tervention on cognition, not mentioning the positive impact that cognitive stimulation may have on the performance of activities of daily living for the elderly. Several studies have addressed the issue of the transfer effect, claiming that cognitive stimulation does not generalize beyond the specific tasks that are worked on [59]. The concept of transfer consists in the generalization of the results obtained to other domains of cognition or to tasks that in some way differ from the target of the training program [59]. Therefore, future studies should seek to understand the effect of the combination of cognitive stimulation and physical activity in the performance of daily living activities in the elderly.”
- Control group information should be included in Table 2
Thanks for this suggestion.
In order to better clarify this issue, we added a column entitled "Comparason groups" to Table 2, with information about the control group. In this sense, we indicate whether the study had a passive control group and how comparisons between groups were carried out.
- I was not sold on the technological aspect of the study as a broad concept; for example, there has been a lot of recent work on step-counting devices etc. which was not covered. Perhaps the term technology could be replaced with exergaming?
Thank for this suggestion. In several paragraphs the term "new technologies" has been equated to or replaced by “exergaming” and “computer programs”, as these are the main forms of new technologies explored in the revised studies.
- Final letters of the title are cropped off.
Thanks for this repair. The word Study in the title was corrected.
- Inclusion of journal rules as part of the introduction were probably not supposed to be included (lines 107+).
We thank the reviewer for this suggestion. The information on those lines has been deleted, as suggested. This information was the journal's guidelines for this section and does not form part of the manuscript's content.
Thank you very much in advance for considering our paper for re-evaluation and I remain available to answer any further comments or concerns.
Sincerely,
Rute Rocha

Reviewer 3 Report
In the article "Importance of technology in the combined interventions of cognitive stimulation and physical activity in the elderly: A Systematic Review Study" Rocha et al. proposes a systematic review that evaluates the randomized controlled studies that involve the role of a double cognitive and physical intervention on the cognitive function of healthy adults. In the context created in developed countries by demographic aging, maintaining the physical and mental health of the elderly is a priority. The article includes the study diagram and 2 tables, the first presenting the critical appraisal of studies included, and the second a summary of the evaluated studies. This second table includes details about the type and frequency of the intervention, about the method and results, data that make it easy to navigate and clear even if it includes a huge number of studies. The approach is unique, including a focus on modern technologies (including video Nintendo and Wii Games) in addition to more traditional methods. The study also mentions the causes that do not make a meta-analysis feasible and exposes the limits of the evaluation. I appreciate this chapter and the mention of new perspectives and the proposals for stratification of some future studies considering in the evaluation/inclusion of baseline cognitive levels. For the effort involved, the original idea and the fluency of the presentation, I propose the publication of the Systematic Review Study.
Author Response
Dear Reviewer,
We would like to thank all the comments made to our article entitled "Importance of technology in the combined interventions of cognitive stimulation and physical activity in cognitive function in the elderly: A Systematic Review Study", by Rute Rocha, Sara M. Fernandes and Isabel M .Santos.
We greatly appreciate the recognition of the innovative character of the study in question. The study of effective strategies to promote active and healthy aging is something that motivates us a lot. Portugal is an aging country, whose challenges associated with aging will increase in the coming decades. Given this reality, there is an urgent need for a growing effort to study this phenomenon and to contribute to the development of strategies that promote health, well-being and quality of life in the elderly.
Thank you so much for suggesting our article for publication and I remain at your disposal for any comments or concerns.
Sincerely,
Rute Rocha

Round 2
Reviewer 2 Report
The authors have dealt with my suggestions and those of the other reviewer and the current article represents a good contribution to the field.